# The Impaired Elasticity of Large Arteries in Systemic Sclerosis Patients

**DOI:** 10.3390/jcm11123256

**Published:** 2022-06-07

**Authors:** Michele Colaci, Luca Zanoli, Alberto Lo Gullo, Domenico Sambataro, Gianluca Sambataro, Maria Letizia Aprile, Pietro Castellino, Lorenzo Malatino

**Affiliations:** 1Rheumatology Clinic, Internal Medicine Unit, AOE Cannizzaro, 95126 Catania, Italy; marialetizia_aprile@libero.it (M.L.A.); malatino@unict.it (L.M.); 2Department of Clinical and Experimental Medicine, University of Catania, 95100 Catania, Italy; dott.zanoli@gmail.com (L.Z.); d.sambataro@hotmail.it (D.S.); dottorsambataro@gmail.com (G.S.); pcastell@unict.it (P.C.); 3Internal Medicine Unit, Policlinico Rodolico—S. Marco, 95123 Catania, Italy; 4Rheumatology Unit, ARNAS Garibaldi, 95123 Catania, Italy; albertologullo@virgilio.it

**Keywords:** pulse wave velocity, carotid distensibility, carotid strain, Young’s elastic modulus, carotid compliance, systemic sclerosis, scleroderma, arterial stiffness

## Abstract

(1) Background: Systemic sclerosis (SSc) is an autoimmune disease characterized by endothelial dysfunction and fibrosis of skin and visceral organs. In the last decade, attention has been focused on the macrovascular involvement of the disease. In particular, the observation of increased arterial stiffness represented an interesting aspect of the disease, as predictor of cardiovascular risk. (2) Methods: We recruited 60 SSc patients (52 ± 12 years old, 90% females) and 150 age/sex-matched healthy controls in order to evaluate both intima-media thickness of the right common carotid artery and arterial stiffness using the B-mode echography and the SphygmoCor system^®^ tonometer. (3) Results: The carotid-femoral pulse wave velocity (PWV) was higher in SSc patients than in controls (8.6 ± 1.7 vs. 7.8 ± 1.5 m/s; *p* < 0.001), as was the carotid-radial PWV (7.8 ± 1.1 vs. 6.7 ± 1.4 m/s; *p* < 0.001). The intima-media thickness was higher in SSc than in controls (654 ± 108 vs. 602 ± 118 µm; *p* = 0.004). The other parameters measured at carotid (radial strain, Young’s modulus, compliance and distensibility) all indicated that arterial stiffness in tension was more pronounced in SSc. Of interest, the direct correlation between PWV and age corresponded closely in SSc. Moreover, a significant difference between SSc and controls as regards the carotid parameters was evident in younger subjects. (4) Conclusions: SSc patients showed an increased arterial stiffness compared to healthy controls. In particular, an SSc-related pathologic effect was suggested by the more pronounced increase in PWV with age and lower values of carotid elasticity in younger SSc patients than in age-matched controls.

## 1. Introduction

Systemic sclerosis (SSc) is characterized by endothelial dysfunction and diffuse vasculopathy. Traditionally, the vascular involvement of SSc was considered mainly microvascular, leading to tissue ischemia. Raynaud’s phenomenon, digital ulcers, as well as pulmonary arterial hypertension are typical SSc features that pathogenetically identify functional and structural alterations of microvasculature [1,2]. However, in the last decade, increasing evidence that large arteries are also affected was accumulated [3,4,5].

Arterial stiffness is a typical sign of arterial dysfunction, even in the absence of overt vascular abnormalities. The main pathogenetic feature is damage to elastin fibers of the aorta and its main branches, which are replaced by collagen overproduction [6]. Furthermore, the increase in vascular smooth-muscle cells and the infiltration of lymphocytes that secrete metalloproteinases and cytokines, such as transforming growth factor-ß, also contribute to arterial wall stiffening [6].

Arterial stiffness is well recognized as an independent cardiovascular risk factor; therefore, its assessment is clinically relevant [7,8,9]. The measurement of arterial stiffness may be easily performed by a simple, non-invasive evaluation of the pulse wave velocity (PWV). Several studies in the literature showed that SSc patients presented increased PWV if compared with healthy age/sex-matched controls, suggesting that the autoimmune disease is also responsible for macrovascular alterations [3,5,10,11,12,13,14,15,16,17,18,19,20,21,22]. Nonetheless, arterial stiffness is a well-recognized crossroad of different pathophysiological states, such as arterial hypertension, diabetes mellitus, smoking-related complications, and dyslipidemia [7,8,9,23]. In this study, we aimed to investigate the macrovascular involvement in SSc by measuring aortic stiffness together with the behavior of the so far unknown carotid elastic parameters, in order to identify additional alterations, if any, of carotid wall elasticity associated with SSc.

## 2. Materials and Methods

### 2.1. Patients

We recruited consecutive SSc patients classified according to the 2013 ACR/EULAR criteria [24] and referred to the Rheumatology Clinic of the Cannizzaro hospital or to the Rheumatology Unit of the ARNAS Garibaldi hospital, both in Catania, Italy. The SSc series was paired with a control group of the same ethnicity, matched for age and sex, randomly selected from a database including healthy outpatients referred to the clinic for cardiovascular diseases prevention, Internal Medicine Unit, Policlinico Rodolico—S.Marco, Catania, Italy.

SSc patients and controls affected by diseases associated with arterial stiffening, such as arterial hypertension, diabetes mellitus, chronic kidney disease, dyslipidemia, atherosclerotic diseases (i.e., myocardial infarction, stroke and peripheral arterial diseases), chronic heart failure and current or former smoking habits, permanent or paroxysmal atrial fibrillation, were excluded from this study.

Chronic vasoactive treatments were not interrupted for the study. However, in the case of prostanoid infusion therapy, the measurements of vascular parameters were performed at least 3 weeks after the last prostanoid administration.

All demographic, clinical, laboratory and instrumental characteristics of SSc patients had already been collected in their clinical records. These records included: clinical history, data from the physical examination, C-reactive protein, erythrocyte sedimentation rate, complete blood counts, indices of liver and renal function, autoantibody profiles, plasma NT-pro-BNP, spirometry, DLCO measurement, chest high-resolution computed tomography, ECG and echocardiography.

All patients gave their informed consent to the study, which was carried out in accordance with the ethical standards of the 1964 Helsinki Declaration and its later amendments and approved by the Ethics Committee of Catania 1 (protocol n. 403, 13 March 2017).

### 2.2. Methods

All participants were studied in a vascular clinic by the same trained operator (L.Z.) blinded to patients’ clinical histories. Vascular evaluations were performed in a quiet room with a controlled temperature of 22 ± 1 °C between 9:00 and 11:00 a.m. The patients were fasting and refrained from caffeine, alcohol and exercise before the study for at least 12 h.

After 15 min of rest in a supine position, brachial blood pressure was measured three times, 2 min apart, using a validated oscillometric device (Spacelabs 90217 ambulatory blood pressure monitor: Issaquah, WA, USA) [25]. The mean value of three measurements was used in this study.

The study of right common carotid artery was performed as previously reported [26]. Longitudinal B-mode (60 Hz, 128 radiofrequency lines) and fast B-mode (600 Hz, 14 radiofrequency lines) images of the artery 2 cm below the carotid bulb were obtained using a high-precision echo tracking device (MyLab One; Esaote, Maastricht, The Netherlands) equipped with a high-resolution (13 MHz) linear-array transducer. The systolic and diastolic internal diameters (D_s_ and D_d_) and the intima-media thickness (IMT) were measured at the right common carotid artery, according to Laurent et al. [27]. It is not excluded that IMT may be slightly different for measurements at the left side.

The right arm radial pulse wave profile was recorded by applanation tonometry (SphygmoCor system^®^, AtCor Medical, Sydney, Australia) after recalibration with brachial mean blood pressure (MBP) and diastolic blood pressure (DBP) in the contralateral arm and was used to calculate carotid pulse pressure (PP). Brachial MBP was calculated as brachial DBP + 1/3 × brachial PP.

The carotid PP was used for the calculation of carotid stiffness indexes [28].

The carotid-femoral and the carotid-radial PWV (cfPWV and crPWV, respectively) were measured with the SphygmoCor device, as previously reported [27], using the foot-to-foot velocity method, the intersecting tangent algorithm and the direct distance between the measurement sites [29]: cfPWV (m/s) = 0.8 × [carotid-femoral direct distance (m)/Δt]; crPWV (m/s) = 0.8 × [carotid-radial direct distance (m)/Δt]. The mean value of two consecutive recordings was used for this analysis. When the difference between the two measurements was ≥0.5 m/s, a third recording was performed, and the median value was used. The augmentation index (AIx%), an indirect measure of arterial stiffness, was calculated as the difference between the late systolic peak and the early systolic peak pressure, divided by the former.

The carotid distensibility, defined as the relative change in luminal area (∆A) during systole for a given pressure change, was calculated as previously described [26], assuming the lumen to be circular and using the following equation: carotid distensibility = ∆A/A × carotid PP.

The carotid strain, defined as the relative change in the vessel diameter during systole, was calculated using the following equation: Strain = (Ds − Dd)/Dd, where Ds is the systolic internal diameter and Dd is the diastolic internal diameter.

The circumferential wall stress (CWS), which represents the tangential force that enlarges the vessel, was calculated as follows: CWS (kPa) = (mean BP × Dm)/2IMTm (where Dm and IMTm were the mean values of the internal diameter and the wall thickness during the cardiac cycle). The cross-sectional compliance coefficient (CC) represents the absolute change in lumen area during systole for a given pressure change and was calculated as follows: CC = stroke change in lumen area/local pulse pressure.

The incremental young elastic modulus (Einc), which represents the elastic properties of the material of the arterial wall (assuming that the vessel wall consists of a homogeneous material), was calculated as previously described [26], using the following equation: Einc = [3(1 + A/WCSA)]/DC, given that WCSA is the mean intima-media cross-sectional area.

### 2.3. Statistical Analysis

All continuous variables are presented as mean ± standard deviation (SD), after confirming their normal distribution by means of the Kolmogorov–Smirnov test; categorical variables are presented as a percentage value.

Clinical and hemodynamic variables were compared using analysis of variance (ANOVA) for continuous variables. Spearman linear regression analysis was performed to verify the existence of any significant correlation between two quantitative variables. *p* values < 0.05 were considered statistically significant.

The statistical analysis was performed using NCSS 2007 and PASS 11 software (Gerry Hintze, Kaysville, UT, USA).

## 3. Results

This cross-sectional study included 60 SSc patients and 150 age/sex-matched healthy controls. Table 1 shows demographic characteristics and findings obtained in this study.

Fifty-six out of 60 (93.3%) SSc patients showed the limited skin subset, 27 patients (45%) presented digital ulcers in their clinical history, 32 (53.3%) interstitial lung disease and 3 (5%) pulmonary arterial hypertension (PAH). Furthermore, anti-centromere or anti-Scl70 autoantibodies were found in 35 (58.3%) and 22 (36.7%) SSc patients, respectively.

All patients were treated with calcium channel blockers for Raynaud’s phenomenon. Moreover, 13 patients used endothelin-1 receptor antagonists for digital ulcer prevention or PAH, while 35 were treated with monthly infusion of prostanoids.

No significant difference between SSc patients and controls as regards the body mass index (BMI) was noted. The values for blood pressure confirmed that no patients with arterial hypertension were included.

The evaluation of the right common carotid artery showed an increase in IMT in SSc patients compared with controls, but a similar internal diameter. Moreover, stiffness of the arterial wall was significantly increased in SSc patients compared with controls (Table 1). In particular, PWV (measured both as carotid-femoral PWV and radial-femoral PWV) was clearly higher in SSc patients than in controls, indicating a diffuse vascular stiffening involving both elastic and muscular arteries in SSc patients.

Carotid-femoral PWV > 10 m/s was found in 13 out of 60 SSc patients and 11 out of 150 controls (21.7% vs. 7.3%; *p* = 0.003). These 13 SSc patients had a mean age of 63.9 (range 52–73) years, whereas the mean age of the entire SSc group was 52 ± 12 years (*p* = 0.0014). No other SSc characteristics distinguished this patient subgroup from the others.

Interestingly, PWV was directly and more closely correlated with age in SSc patients compared with healthy controls, with a steeper regression line, suggesting that SSc is responsible for an accelerated loss of elastic properties of large vessels (Figure 1).

Macrovascular stiffness was consistent with structural changes of the arterial wall, as shown by the higher IMT of the common carotid artery in SSc than in controls (Table 1, Figure 2), although a similar pattern of direct association between PWV and IMT was demonstrated in both SSc patients and healthy subjects (Figure 3).

An increased carotid elastic modulus index, along with reduced radial strain, compliance and distensibility, were also observed in SSc patients, compared with controls (Table 1). As shown in Figure 4, in younger SSc patients, carotid wall presented a reduction in its elasticity in comparison with healthy individuals. These findings suggest a direct effect of SSc on arterial wall, leading to a precocious arterial aging. In fact, carotid elastic changes (compliance, distensibility, and radial strain) in sclerodermic patients tends to be more different from controls in younger subjects and overlapping in the elderly (Figure 4).

Vasoactive therapies did not seem to influence our findings. Finally, except for SSc patients’ age, no significant correlations between SSc characteristics and vascular parameters were found.

## 4. Discussion

In this study, we evaluated the arterial stiffness of large vessels in a group of SSc patients compared with healthy controls. PWV, an index of aortic stiffness, was higher in SSc patients than in controls, in accordance with the literature [3,4,5,6,10,11,12,13,14,15,16,17,18,19,20,21,22]. In particular, cfPWV was directly and more closely correlated with age in patients than in controls, and was positively associated with IMT. We also demonstrated that vascular stiffening involved both muscular and elastic arteries of SSc patients, as shown by the higher values for both crPWV and cfPWV (Table 1).

In a previous study [18], a cutoff of 9 m/s for cfPWV was considered in order to identify patients with a significant increase. At variance, in the present study, a cutoff of 10 m/s was considered, according to the ESC guidelines for arterial hypertension [30]. However, it would be more correct to consider the absolute values of PWV without choosing a cutoff, because no data are so far available on the clinical and prognostic significance of individual PWV values in rheumatic diseases.

Of note, the increase in carotid Young’s elastic modulus and the reduction in radial strain, compliance and distensibility, provided for the first time in SSc evidence further corroborating the concept that a complex dysfunction of macrovascular arterial bed occurs in SSc patients. The carotid stiffening was particularly evident in younger SSc patients, thus emphasizing its association with SSc.

Vascular dysfunction of SSc involved both elastic (i.e., carotid artery and aorta) and muscular (i.e., brachial artery) arteries. Moreover, in SSc patients we found the coexistence of stiffening and thickening processes. The increase in the Young’s elastic modulus suggested that alterations of the bioelastic material occur in the arterial wall in SSc. As a consequence of the stiffening process, the radial strain and the distensibility of the common carotid artery were reduced. Of interest, despite the increased common carotid artery IMT in SSc compared with control subjects, the internal diameter was comparable, thus suggesting that, in SSc patients, the thickening of large elastic arteries may proceed in parallel with the enlargement of the arterial wall. Further studies may be needed to confirm this hypothesis.

The increase in arterial stiffness in SSc is a crucial point in the assessment of the cardiovascular risk in SSc patients [5,10,12], providing additional predictive value above IMT measurement for the association with high risk of cardiovascular disease [31]. In our previous study [32] including female patients affected by SSc or diabetes mellitus, the incidence of established coronary artery disease was lower in SSc patients than in diabetics, but similar between the two groups when subjects older than 65 years were considered. This would mean that, when SSc patients grow older than 65 years, the SSc phenotype may become equivalent to that of diabetes.

Rheumatic diseases, such as rheumatoid arthritis (RA), were widely studied as independent risk factors for the increase in cardiovascular risk, due to the chronic inflammatory state [33,34,35]. In the literature, increased incidence of major adverse cardiovascular events was largely demonstrated for RA patients, in association with long disease duration and scarce disease control [33,34,35]. Moreover, it is well known that the chronic inflammatory state leads to endothelial activation/lesion and surface expression of adhesion molecules for migration of leukocytes [36]. The inflammatory infiltration of atherosclerotic plaques and progression of vascular injury through the inflammatory pathway may be considered as the *primum movens* of premature atherosclerosis in RA patients [33,34,35,36].

Differently from other rheumatic diseases, SSc is characterized by endothelial dysfunction even from its early phase [1,2]. Raynaud’s phenomenon, or the onset of digital ulcers, may be considered a direct consequence of the imbalance between endothelium-derived relaxation and vasoconstrictive factors. Endothelial dysfunction is considered an early marker of atherosclerosis [37], so, therefore, we may assume that SSc could contribute *per se* to the development of accelerated atherosclerosis. Consistently, our findings (Figure 1 and Figure 4) suggested that SSc could be involved in the pathogenetic chain of arterial stiffening. In this respect, in a previous study [38], we raised a suggestive hypothesis regarding aortic wall damage in SSc patients. In particular, we found a significant association between nailfold videocapillaroscopy abnormalities and aortic root dilation. These findings could lead us to hypothesize that a microvascular dysfunction of the aortic *vasa vasorum* may contribute to the early damage of the aortic wall. Then, fibrosis, due to collagen overproduction replacing elastin fibers, completes the aortic remodeling. The absence of histological studies on aortic wall from SSc patients, unfortunately, so far does not allow confirmation of this hypothesis.

Overall, the presence of a macrovascular involvement in SSc is widely accepted, even though the large heterogeneity of the methodologies used in the literature has so far raised some inconsistencies [3]. However, the presence of SSc-specific endothelial dysfunction makes plausible the development of an SSc-related macrovascular alteration.

In clinical practice, SSc patients may be affected also by systemic hypertension, diabetes, and dyslipidemia or could be smokers. Therefore, the coexistence of several cardiovascular risk factors makes the identification of the intrinsic role of SSc more difficult. For this reason, in this study, we focused on SSc patients without other conditions known to be traditional cardiovascular risk factors.

In our study, we did not find correlations between arterial stiffness parameters and SSc patients’ features, besides patients’ age. It is probable that the relatively low number of cases included in our study (i.e., few cases with diffuse skin subset or PAH) did not allow us to find specific correlations found in previous studies [12,39]. Further larger cohort studies are needed to clarify this issue.

In conclusion, the evaluation of carotid elasticity, carried out for the first time in our study, could facilitate a better understanding of the specific characteristics of macrovascular abnormalities in patients with SSc.

This study, however, has limitations. In fact, some unknown or underestimated factors might have influenced our findings. For instance, many drugs used in SSc are vasoactive substances that could counteract the long-term evolution of arterial stiffness. Furthermore, immunosuppressive agents may inhibit leukocyte activities also in the arterial wall, thus influencing the natural progression of disease. Therefore, considering the high number of confounders influencing endothelial function and arterial alterations, protocols for the evaluation of cardiovascular risk in SSc should be designed within very large, multicenter, prospective studies.

## Figures and Tables

**Figure 1 jcm-11-03256-f001:**
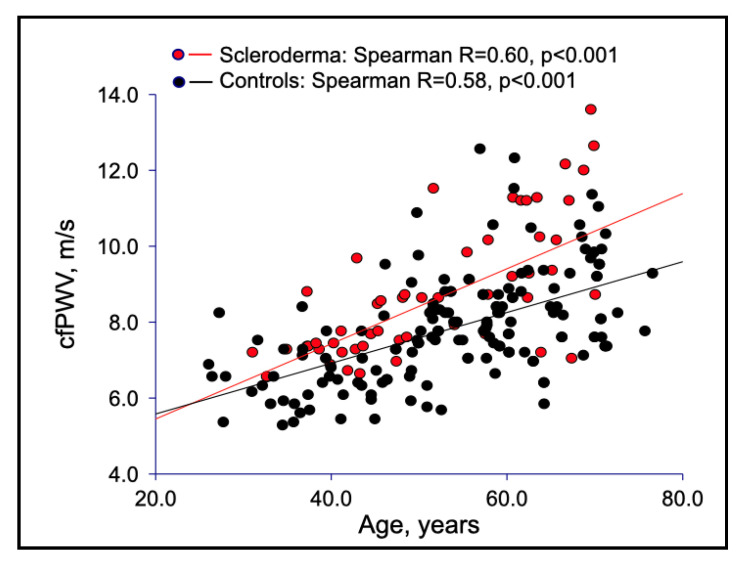
Correlations between carotid femoral PWV and age in SSc patients versus healthy controls.

**Figure 2 jcm-11-03256-f002:**
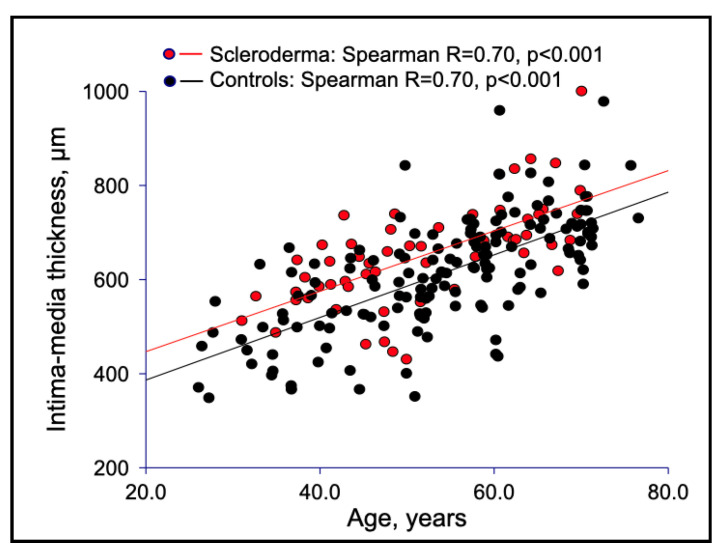
Correlations between IMT and age in SSc patients versus healthy controls.

**Figure 3 jcm-11-03256-f003:**
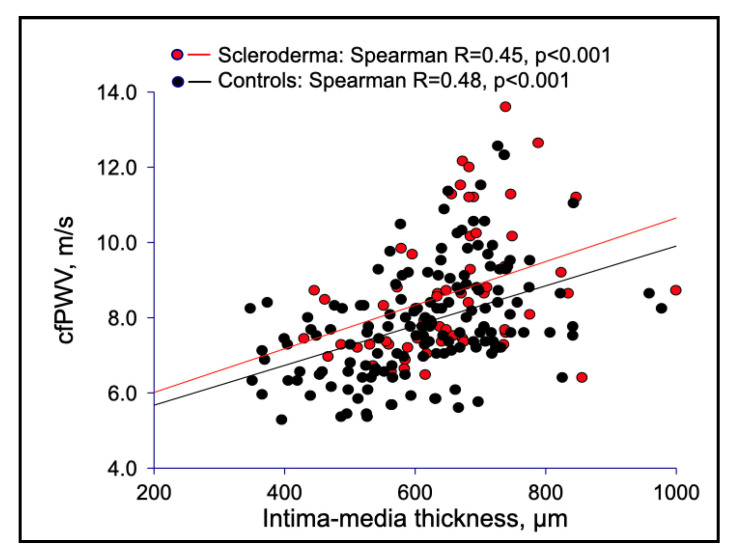
Correlations between carotid femoral PWV and IMT in SSc patients versus healthy controls.

**Figure 4 jcm-11-03256-f004:**
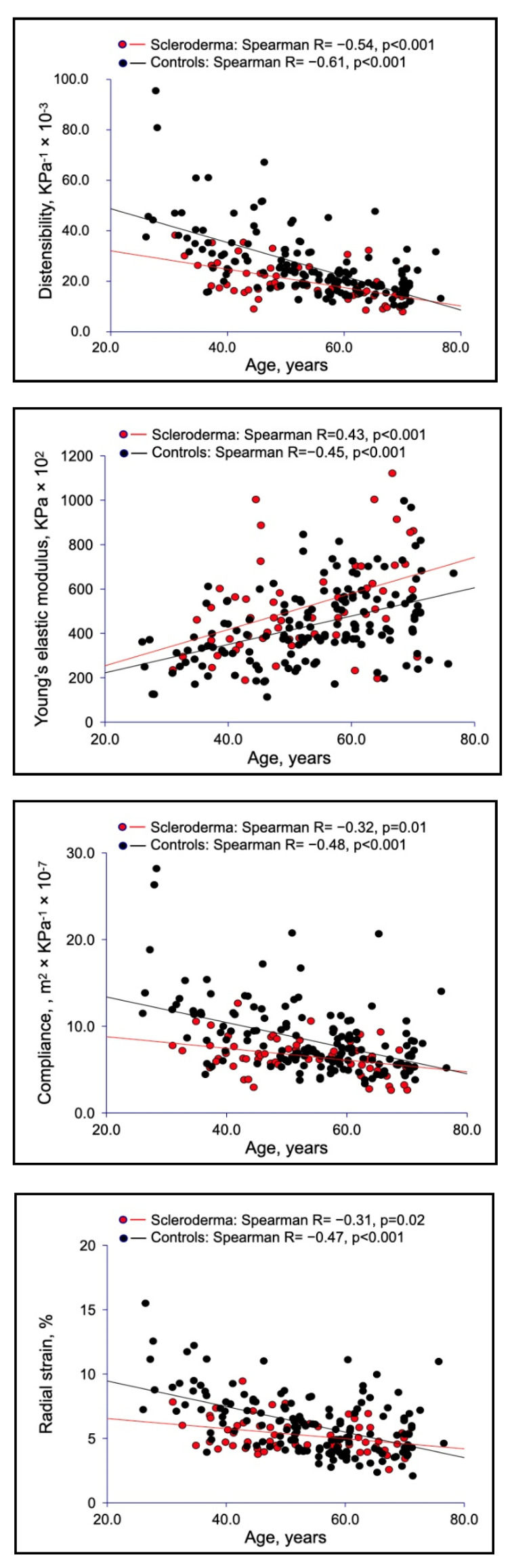
Carotid wall elasticity evaluation in SSc patients versus healthy controls.

**Table 1 jcm-11-03256-t001:** Data obtained in SSc patients and age/sex-matched healthy controls.

Parameter	SSc Patients *n* = 60	Controls *n* = 150	*p*-Values
Age, years	52 (12)	53 (13)	0.93
Females, %	90	86	0.62
BMI, Kg/m^2^	24.4 (3.7)	25.0 (4.3)	0.40
SBP, mmHg	124 (19)	121 (18)	0.45
DBP, mmHg	70 (9)	71 (9)	0.62
MBP, mmHg	90 (12)	67 (11)	0.81
HR, b/m	68 (9)	67 (11)	0.46
cfPWV, m/s	8.6 (1.7)	7.8 (1.5)	**<0.001**
crPWV, m/s	7.8 (1.1)	6.7 (1.4)	**<0.001**
AIx%	33 (11)	29 (14)	0.06
Internal diameter, mm	5.78 (0.69)	5.69 (0.69)	0.39
Intima-media thickness, µm	654 (108)	602 (118)	**0.004**
Radial strain, %	5.28 (1.29)	6.14 (2.32)	**0.01**
Distensibility, KPa^−1^ × 10^−3^	20.3 (7.5)	26.4 (13.5)	**0.001**
Young’s elastic modulus, KPa × 10^2^	517 (210)	436 (178)	**0.01**
Circumferential wall stress, KPa	54 (13)	58 (15)	0.07
CC, m^2^ × KPa^−1^ × 10^−7^	6.72 (1.95)	7.98 (3.31)	**0.02**

Data are shown as mean (standard deviation). SSc = systemic sclerosis; BMI = body mass index; SBP = systolic blood pressure; DBP = diastolic blood pressure; MBP = mean blood pressure; HR = heart rate; cfPWV = carotid-femoral pulse wave velocity; crPWV = carotid-radial pulse wave velocity; AIx% = augmentation index; CC = cross-sectional compliance coefficient. The internal diameter was measured 2 cm below the bulb of the right common carotid artery. Significant *p* values have been reported in bold.

## Data Availability

The data presented in this study are available on request from the corresponding author. The data are not publicly available due to privacy restrictions.

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
