# Peer review of "The Impaired Elasticity of Large Arteries in Systemic Sclerosis Patients"

_jcm, 2022, doi:10.3390/jcm11123256_

Round 1

Reviewer 1 Report

1) Were patients with mixed connective tissue disease but fulfilling the SSc EULAR/ACR criteria also included or were only patients with just SSc included?

2) Other atherosclerotic arterial diseases like myocardial infarction or peripheral artery disease are also commonly associated with a higher arterial stiffness, while within the methods section only previous stroke was an exclusion criteria. Why did the authors not exclude other forms of atherosclerotic diseases?

3) Why was only the IMT of the right common carotid artery measured and not the mean value of both common carotid arteries?

4) I have doubts that a measurement of the IMT was so exact and possible that the results could be given in µm, even with a high resolution echo device. Did the operator the measurements manually or was it automated by the software?

5) How many SSc patients had already vascular complications like digital ulcers or pulmonary hypertension? There are investigations that patients with such complications may have a higher arterial stiffness. Additionally, how many SSc patients had the diffuse form and the limited form of SSc as also the SSc subtype may affect arterial stiffness? Could you also present data between both SSc subtypes if there are differences of the investigated parameters of arterial stiffness?

6) According to the ESC guidelines for arterial hypertension (Williams B, et al. European Heart Journal (2018) 39, 3021–3104), a PWV of > 10m/s is considered as pathologic. How many of your patients had such a pathologic PWV? Although the authors stated in the discussion that a cut-off value was not chosen, it would be interesting how many of the measured patients had a significant PWV.

7) One large clinical aspect is missing, namely if there are any implications of the investigated parameters with clinical parameters of SSc. Do the measured parameters of arterial stiffness correlate with clinical or laboratory parameters like modified Rodnan skin score, EUSTAR index, capillary patterns, GFR, CRP, ...? As a physician, it would be, for example, of interest if I need to screen SSc patients for their arterial stiffness due to potential progression of a renal involvement.

8) As already stated by the authors, vasoactive substances and drugs may affect arterial stiffness. How many of the patients had such a therapy?

Author Response

We would like to thank the reviewer for his careful work which has allowed us to improve our study.

Reviewer 2 Report

The authors reported the arterial stiffness of large vessels in a group of systemic sclerosis (SSc) patients compared with healthy controls and found PWV and IMT was higher in SSc patients than controls. In particular, cfPWV and crPWV was directly and more closely correlated with age in patients than in controls, and resulted positively associated with IMT.

However, some concerns have been raised.

1.     As the authors noted several studies in the literature showed that SSc patients presented increased PWV if compared with healthy age/sex-matched controls, suggesting that the autoimmune disease is also responsible for macrovascular alterations [3,5,10-22]. How the novel finding of this study?

2.     What the novel factors that affect the different parameters of arterial stiffness of large vessels (diastolic and systolic internal diameters (Ds andDd), the intima-media thickness (IMT), carotid pulse pressure (PP), carotid stiffness indexes, carotid-femoral and the carotid-radial PWV, carotid distensibility, carotid strain, circumferential wall stress, the incremental young elastic modulus) in SSc patients?

3.     Why use analysis of variance (ANOVA) for continuous variables in only two groups?

4.     Lack of parameters to prove this finding in SSc patients, such as inflammatory markers, endothelial dysfunction etc in this study.

5.     In this study excluded systemic hypertension, diabetes, dyslipidemia, smokers and can not the same as the real world practice of SSc patients.

Author Response

We would like to thank the reviewer for his revision which has allowed us to improve our study.

Round 2

Reviewer 1 Report

I have no further comments